# Impact of Smoking Status on Mortality in STEMI Patients Undergoing Mechanical Reperfusion for STEMI: Insights from the ISACS–STEMI COVID-19 Registry

**DOI:** 10.3390/jcm11226722

**Published:** 2022-11-13

**Authors:** Giuseppe De Luca, Magdy Algowhary, Berat Uguz, Dinaldo C. Oliveira, Vladimir Ganyukov, Zan Zimbakov, Miha Cercek, Lisette Okkels Jensen, Poay Huan Loh, Lucian Calmac, Gerard Roura i Ferrer, Alexandre Quadros, Marek Milewski, Fortunato Scotto D’Uccio, Clemens von Birgelen, Francesco Versaci, Jurrien Ten Berg, Gianni Casella, Aaron Wong Sung Lung, Petr Kala, José Luis Díez Gil, Xavier Carrillo, Maurits Dirksen, Victor M. Becerra-Munoz, Michael Kang-yin Lee, Dafsah Arifa Juzar, Rodrigo de Moura Joaquim, Roberto Paladino, Davor Milicic, Periklis Davlouros, Nikola Bakraceski, Filippo Zilio, Luca Donazzan, Adriaan Kraaijeveld, Gennaro Galasso, Lux Arpad, Marinucci Lucia, Guiducci Vincenzo, Maurizio Menichelli, Alessandra Scoccia, Aylin Hatice Yamac, Kadir Ugur Mert, Xacobe Flores Rios, Tomas Kovarnik, Michal Kidawa, Josè Moreu, Flavien Vincent, Enrico Fabris, Iñigo Lozano Martínez-Luengas, Marco Boccalatte, Francisco Bosa Ojeda, Carlos Arellano-Serrano, Gianluca Caiazzo, Giuseppe Cirrincione, Hsien-Li Kao, Juan Sanchis Forés, Luigi Vignali, Helder Pereira, Stephane Manzo, Santiago Ordoñez, Alev Arat Özkan, Bruno Scheller, Heidi Lehtola, Rui Teles, Christos Mantis, Ylitalo Antti, João António Brum Silveira, Rodrigo Zoni, Ivan Bessonov, Stefano Savonitto, George Kochiadakis, Dimitrios Alexopulos, Carlos E. Uribe, John Kanakakis, Benjamin Faurie, Gabriele Gabrielli, Alejandro Gutierrez Barrios, Juan Pablo Bachini, Alex Rocha, Frankie Chor-Cheung Tam, Alfredo Rodriguez, Antonia Anna Lukito, Veauthyelau Saint-Joy, Gustavo Pessah, Andrea Tuccillo, Giuliana Cortese, Guido Parodi, Mohamed Abed Bouraghda, Elvin Kedhi, Pablo Lamelas, Harry Suryapranata, Matteo Nardin, Monica Verdoia

**Affiliations:** 1Division of Clinical and Experimental Cardiology, AOU Sassari, University of Sassari, 07100 Sassari, Italy; 2Division of Cardiology, Assiut University Heart Hospital, Assiut University, Asyut 71511, Egypt; 3Division of Cardiology, Bursa City Hospital, 16000 Bursa, Turkey; 4Pronto de Socorro Cardiologico “ Prof. Luis Tavares”, Centro PROCAPE, Federal University of Pernambuco, Recife 50000-000, Brazil; 5Department of Heart and Vascular Surgery, State Research Institute for Complex Issues of Cardiovascular Diseases, 650000 Kemerovo, Russia; 6University Clinic for Cardiology, Medical Faculty, Ss’ Cyril and Methodius University, 1000 Skopje, North Macedonia; 7Centre for Intensive Internal Medicine, University Medical Centre, 1000 Ljubljana, Slovenia; 8Division of Cardiology, Odense Universitets Hospital, 5000 Odense, Denmark; 9Department of Cardiology, National University Hospital, Singapore 117597, Singapore; 10Clinic Emergency Hospital of Bucharest, 010001 Bucharest, Romania; 11Interventional Cardiology Unit, Heart Disease Institute, Hospital Universitari de Bellvitge, 08016 Barcelona, Spain; 12Instituto de Cardiologia do Rio Grande do Sul, Porto Alegre 90000-00, Brazil; 13Division of Cardiology, Medical University of Silezia, 40-002 Katowice, Poland; 14Division of Cardiology, Ospedale del Mare, 00156 Napoli, Italy; 15Department of Cardiology, Medisch Spectrum Twente, Thoraxcentrum Twente, 7541 Enschede, The Netherlands; 16Division of Cardiology, Ospedale Santa Maria Goretti Latina, 04100 Latina, Italy; 17Division of Cardiology, St Antonius Hospital, 3434 Nieuwegein, The Netherlands; 18Division of Cardiology, Ospedale Maggiore Bologna, 40100 Bologna, Italy; 19Department of Cardiology, National Heart Center, Singapore 169609, Singapore; 20University Hospital Brno, Medical Faculty of Masaryk, University Brno, 60200 Brno, Czech Republic; 21H. Universitario y Politécnico La Fe, 46001 Valencia, Spain; 22Cardiology, Hospital Germans Triasi Pujol, 8918 Badalona, Spain; 23Division of Cardiology, Northwest Clinics, 1811 Alkmaar, The Netherlands; 24Cardiology, Hospital Clínico Universitario Virgen de la Victoria, 29000 Malaga, Spain; 25Department of Cardiology, Queen Elizabeth Hospital, University of Hong Kong, Hong Kong 999077, China; 26Department of Cardiology and Vascular Medicine, National Cardiovascular Center “Harapan Kita”, University of Indonesia, Jakarta 11420, Indonesia; 27Instituto de Cardiologia de Santa Catarina Praia Comprida, Sao Jose 88100-000, Brazil; 28Division of Cardiology, Clinica Villa dei Fiori, 80011 Acerra, Italy; 29Department of Cardiology, University Hospital Centre, University of Zagreb, 10000 Zagreb, Croatia; 30Invasive Cardiology and Congenital Heart Disease, Patras University Hospital, 26221 Patras, Greece; 31Center for Cardiovascular Diseases, 6000 Ohrid, North Macedonia; 32Division of Cardiology, Ospedale Santa Chiara di Trento, 38014 Trento, Italy; 33Division of Cardiology, Ospedale “S. Maurizio”, 39100 Bolzano, Italy; 34Division of Cardiology, UMC Utrecht, 3584 CX Utrecht, The Netherlands; 35Division of Cardiology, Ospedale San Giovanni di Dio e Ruggi d’Aragona, 84070 Salerno, Italy; 36Cardiology, Maastricht University Medical Center (UMC+), 6229 Maastricht, The Netherlands; 37Division of Cardiology, Azienda Ospedaliera “Ospedali Riuniti Marche Nord”, 61121 Pesaro, Italy; 38Division of Cardiology, AUSL-IRCCS Reggio Emilia, 42121 Reggio Emilia, Italy; 39Division of Cardiology, Ospedale “F. Spaziani”, 03100 Frosinone, Italy; 40Division of Cardiology, Ospedale “Sant’Anna”, 44121 Ferrara, Italy; 41Department of Cardiology, Bezmialem Vakif University Hospital, 34093 Istanbul, Turkey; 42Division of Cardiology, Faculty of Medicine, Eskisehir Osmangazi University, 02640 Eskisehir, Turkey; 43Cardiology, Complexo Hospetaliero Universitario La Coruna, 15001 La Coruna, Spain; 44Cardiology University Hospital Prague, 12808 Prague, Czech Republic; 45Central Hospital of Medical University of Lodz, 90-008 Lodz, Poland; 46Division of Cardiology, Complejo Hospitalario de Toledo, 45001 Toledo, Spain; 47Division of Cardiology, Center Hospitalier Universitaire de Lille, 59000 Lille, France; 48Azienda Ospedaliero—Universitaria Ospedali Riuniti, 34142 Trieste, Italy; 49Division of Cardiology, Hospital Cabueñes, 33201 Gijon, Spain; 50Division of Cardiology, Ospedale Santa Maria delle Grazie, 80078 Pozzuoli, Italy; 51Division of Cardiology, Hospital Universitario de Canarias, 38001 Santa Cruz de Tenerife, Spain; 52Division of Cardiology, Hospital Puerta de Hierro Majadahonda, 28222 Madrid, Spain; 53Division of Cardiology, Ospedale “G Moscati”, 81031 Aversa, Italy; 54Division of Cardiology, Ospedale Civico Arnas, 90100 Palermo, Italy; 55Cardiology Division, Department of Internal Medicine, National Taiwan University Hospital, Tapei 8865600, Taiwan; 56Division of Cardiology, Hospital Clinico Universitario de Valencia, 46010 Valencia, Spain; 57Interventional Cardiology Unit, Azienda Ospedaliera Sanitaria, 43121 Parma, Italy; 58Cardiology Department, Hospital Garcia de Orta (Pragal), 2805-267 Almada, Portugal; 59Division of Cardiology, CHU Lariboisière, AP-HP, INSERM UMRS 942, Paris VII University, 75010 Paris, France; 60Instituto Cardiovascular de Buenos Aires, Buenos Aires 1428, Argentina; 61Cardiology Institute, Instanbul University, 34000 Instanbul, Turkey; 62Division of Cardiology, Clinical and Experimental Interventional Cardiology, University of Saarland, 66421 Homburg, Germany; 63Division of Cardiology, Oulu University Hospital, 90220 Oulu, Finland; 64Division of Cardiology, Hospital de Santa Cruz, CHLO—Nova Medical School, Centro de Estudos de Doenças Crónicas (CEDOC), 1000 Lisbon, Portugal; 65Division of Cardiology, Kontantopoulion Hospital, 104 31 Athens, Greece; 66Division of Cardiology, Heart Centre Turku, 20521 Turku, Finland; 67Division of Cardiology, Hospital de Santo António, 4099-001 Porto, Portugal; 68Tyumen Cardiology Research Center, 625026 Tyumen, Russia; 69Department of Teaching and Research, Instituto de Cardiología de Corrientes “Juana F. Cabral”, Corrientes W3400CDS, Argentina; 70Division of Cardiology, Ospedale “A. Manzoni”, 23900 Lecco, Italy; 71Cardiology, Iraklion University Hospital, 70001 Crete, Greece; 72Division of Cardiology, Attikon University Hospital, 10431 Athens, Greece; 73Division of Cardiology, Universidad UPB- CES, Medellin 050001, Colombia; 74Division of Cardiology, Alexandra Hospital, 10431 Athens, Greece; 75Division of Cardiology, Groupe Hospitalier Mutualiste de Grenoble, 38000 Grenoble, France; 76Interventional Cardiology Unit, Azienda Ospedaliero Universitaria “Ospedali Riuniti”, 60100 Ancona, Italy; 77Division of Cardiology, Hospital Puerta del Mar, 11001 Cadiz, Spain; 78Instituto de Cardiologia Integral, Montevideo 11700, Uruguay; 79Department of Cardiology and Cardiovascular Interventions, Instituto Nacional de Cirugía Cardíaca, Montevideo 11700, Uruguay; 80Department of Cardiology, Queen Mary Hospital, University of Hong Kong, Hong Kong 999077, China; 81Division of Cardiology, Otamendi Hospital, Buenos Aires 1001, Argentina; 82Cardiovascular Department, Siloam Lippo Village Hospital, Heart Center, Pelita Harapan University, Tangerang 15810, Indonesia; 83Cardiology, Center Hospitalier d’Antibes Juan Les Pins, 06600 Antibes, France; 84Division of Cardiology, Hospiatl Cordoba, Cordoba 5000, Argentina; 85Department of Statistical Sciences, University of Padova, 35121 Padova, Italy; 86Cardiology, Azienda Ospedaliera, 16033 Lavagna, Italy; 87Division of Cardiology, Blida University Hospital, Blida 09000, Algeria; 88Division of Cardiology, Hopital Erasmus, Universitè Libre de Bruxelles, 1070 Bruxelles, Belgium; 89Division of Cardiology, Radboud University Medical Center, 6525 Nijmegen, The Netherlands; 90Department of Internal Medicine, Ospedale Riuniti, 25121 Brescia, Italy; 91Division of Cardiology, Ospedale Degli Infermi, ASL Biella, 13900 Biella, Italy

**Keywords:** myocardial infarction, smoking paradox, percutaneous coronary intervention, COVID-19

## Abstract

The so-called “smoking paradox”, conditioning lower mortality in smokers among STEMI patients, has seldom been addressed in the settings of modern primary PCI protocols. The ISACS–STEMI COVID-19 is a large-scale retrospective multicenter registry addressing in-hospital mortality, reperfusion, and 30-day mortality among primary PCI patients in the era of the COVID-19 pandemic. Among the 16,083 STEMI patients, 6819 (42.3%) patients were active smokers, 2099 (13.1%) previous smokers, and 7165 (44.6%) non-smokers. Despite the impaired preprocedural recanalization (*p* < 0.001), active smokers had a significantly better postprocedural TIMI flow compared with non-smokers (*p* < 0.001); this was confirmed after adjustment for all baseline and procedural confounders, and the propensity score. Active smokers had a significantly lower in-hospital (*p* < 0.001) and 30-day (*p* < 0.001) mortality compared with non-smokers and previous smokers; this was confirmed after adjustment for all baseline and procedural confounders, and the propensity score. In conclusion, in our population, active smoking was significantly associated with improved epicardial recanalization and lower in-hospital and 30-day mortality compared with previous and non-smoking history.

## 1. Introduction

Coronary artery disease still represents the leading cause of mortality in developed countries. While large attention has been paid to the identification of new risk factors [1,2,3,4], traditional risk factors, especially cigarette smoking, cannot be neglected. In fact, still approximately 30% of all deaths due to coronary artery disease (CAD) in the United States annually are attributable to smoking [5]. 

Several studies have been conducted, especially in the setting of ST-segment elevation myocardial infarction (STEMI), suggesting the existence of a “smokers’ paradox,” related to the more favorable outcome of smokers compared with non-smokers [6,7,8,9,10]. Similar findings have been observed among patients with acute ischemic stroke, acute heart failure, and cardiac arrest [11,12,13,14]. 

This paradoxically lower mortality observed among smokers was mainly attributed to their younger age, fewer comorbidities, lesser extent of CAD, in addition to potential pathophysiological differences between smokers and non-smokers, including a greater thrombus burden in smokers, leading to greater efficacy of thrombolytic therapy [15,16,17], and greater responsiveness to antiplatelet therapies [18,19,20,21]. However, primary PCI, when applied in a timely fashion, currently represents the best indicated reperfusion therapy for the treatment of STEMI. Several reports have investigated the prognostic impact of smoking with contrasting results. In the COVID era, the increased susceptibility of smokers to respiratory disease and the enhanced thrombotic risk associated with COVID-19 infection, could influence the existence of different outcome results according to smoking status. Moreover, recent reports have clearly shown a reduction in acute coronary cases during the pandemic, presumably due to a public fear of coronavirus contagion that impacted on patient willingness to present to a hospital [22,23,24,25,26,27]. An additional observation was the prolonged time from symptom onset to treatment [28,29,30] that contributed to explain the higher mortality among STEMI patients observed in 2020. 

Therefore, the aim of the present study was to investigate the impact of smoking status on angiographic and clinical outcome in a large cohort of patients enrolled also during the COVID-19 pandemic.

## 2. Materials and Methods

Our study population is represented by patients enrolled in the International Study on Acute Coronary Syndromes—ST-segment Elevation Myocardial Infarction (ISACS–STEMI) COVID-19, a large-scale retrospective multicenter registry involving primary PCI centers from Europe, Latin America, South-East Asia, and North Africa, including patients treated from the 1st of March until the 30th of June 2019 (pre-COVID period) and from the 1st of March until the 30th of June 2020 (COVID period) who underwent SARS-Cov-2 screening [31]. 

We collected demographic, clinical, procedural data, data on total ischemia time, door-to-balloon time, referral to primary PCI facility, PCI procedural data, in-hospital outcomes, including death, Stent Thrombosis (according to the ARC definition, [32]), and 30-day mortality. The study was approved by the Ethical Committee of AOU Maggiore della Carità, Novara (Protocol 571/CE date of approval 20/05/2020).

Statistical data analysis was performed by the use of SPSS Statistics Software 23.0 (IBM SPSS Inc., Chicago, IL, USA). Quantitative variables were described using median and interquartile range. Absolute frequencies and percentages were used for qualitative variables. ANOVA or the Mann–Whitney and chi-square test were used for continuous and categorical variables, respectively. Normal distribution of continuous variables was tested by the Kolmogorov–Smirnov test). Primary study endpoint was in-hospital mortality. Secondary study endpoints were postprocedural TIMI 3 flow and 30-day mortality. We used the propensity score technique to account for potential confounding between groups, as previously described [33,34]. For each patient, a propensity score indicating the likelihood of being active was calculated through step-forward logistic regression analysis that identified variables independently associated with active smoking. We included baseline clinical variables associated with active smoking at univariable analysis (inclusion in the model: *p* < 0.05; exclusion from the model: *p* < 0.1). The following variables were entered into the model: age, gender, diabetes, hypertension, family history of CAD, previous STEMI, previous PCI, previous CABG, type of referral, ischemia time, door-to-balloon time, anterior STEMI, out-of-hospital cardiac arrest, cardiogenic shock, rescue PCI for failed thrombolysis, in-hospital RASI therapy, COVID positivity, year of revascularization (2019 vs. 2020), radial access, in-stent thrombosis, multivessels, disease, preprocedural TIMI flow, stenting, DES, mechanical support, DAPT, and additional PCI. The stepwise selection of the variable and estimation of significant probabilities were computed by means of maximal likelihood ratio test. The χ2 value was calculated from the log of the ratio of maximal partial likelihood functions. The additional value of each category of variables added sequentially was evaluated on the basis of the increases in the overall likelihood statistic ratio. The final score was built according to the global χ2 value of the multivariate statistical model and the χ2 value of each variable. The discriminatory performance of the propensity score was assessed by the receiver operating characteristic curve method, which indicated a good accuracy of the propensity score model (area under the curve = 0.83) [35].

The consistency of the main results for the primary outcome of the study was investigated according to propensity score values (below and over the median).

Multivariable Cox and logistic regression analyses were performed to identify the impact of smoking on primary and secondary study endpoints after adjustment for baseline con-founding factors between the two groups. All significant variables (set at a *p*-value < 0.1) were entered in block into the model. A *p* < 0.05 was considered statistically significant. The data coordinating center was established at the Eastern Piedmont University, Novara, Italy.

## 3. Results

Our population is represented by 16,083 STEMI patients. A total of 6819 (42.3%) patients were active smokers, 2099 (13.1%) previous smokers, and 7165 (44.6%) non-smokers. As shown in Table 1, active smokers were nine years younger and more often males compared with non-smokers. Smokers were less often affected by diabetes (*p* < 0.001), hypertension (*p* < 0.001), and hypercholesterolemia (*p* < 0.001), with lower prevalence of previous STEMI (*p* < 0.001), previous PCI (*p* < 0.001), or CABG (*p* < 0.001), but more often had a positive family history of CAD (*p* < 0.001). Smokers had a shorter ischemia time (*p* < 0.001), less often had anterior MI (*p* < 0.001), cardiogenic shock (*p* < 0.001), and out-of-hospital cardiac arrest (*p* < 0.001), but more often rescue PCI failed after thrombolysis (*p* < 0.001). Angiographic features are displayed in Table 2. Smokers less often had multivessel disease (*p* < 0.001), in-stent thrombosis (*p* < 0.001), received less often a mechanical support (*p* < 0.001) or underwent additional PCI (*p* = 0.001), whereas they received more often a coronary stent (*p* < 0.001), a DES (*p* < 0.001) and DAPT (*p* < 0.001). Despite the impaired preprocedural recanalization (*p* < 0.001), smokers had a significantly better postprocedural TIMI flow as compared to non-smokers (*p* < 0.001) (Figure 1). A significant association was observed with the percentage of SARS-COV 2 positive patients, and less often observed among smokers. Our main results were confirmed in both pre-COVID (*p* < 0.001) and COVID era (*p* < 0.001), and in both young (*p* < 0.001) and elderly patients (*p* = 0.013) (Appendix A). The results were additionally confirmed in the analysis based on the propensity score (Appendix A). The association between active smoking and postprocedural TIMI flow was confirmed after adjustment for all confounders (age, gender, diabetes, hypertension, hypercholesterolemia, family history of CAD, previous STEMI, previous PCI, previous CABG, access by ambulance, ischemia time, door-to-balloon time, anterior MI, out-of-hospital cardiac arrest, cardiogenic shock, rescue PCI, radial access, anterior MI, infarct-related artery, in-stent thrombosis, preprocedural TIMI flow 0, use of stent and DES, mechanical support, DAPT, multivessel disease, additional PCI, year of intervention, propensity score, and COVID positivity) (OR [95% CI] = 1.18 [1.04–1.36), *p* = 0.014).

Smokers had a significantly lower in-hospital (Figure 2) and 30-day (Figure 3) mortality compared with non-smokers and previous smokers; this was confirmed in both pre-COVID (*p* < 0.001) and COVID era (*p* < 0.001) and in young (*p* < 0.001) and older patients (*p* < 0.001) (Appendix A). The results were additionally confirmed in the analysis based on the propensity score (Appendix A). The association between active smoking and better survival was confirmed at multivariate analysis after adjustment for all confounders (age, gender, diabetes, hypertension, hypercholesterolemia, family history of CAD, previous STEMI, previous PCI, previous CABG, access by ambulance, ischemia time, door-to-balloon time, anterior MI, out-of-hospital cardiac arrest, cardiogenic shock, rescue PCI, radial access, anterior MI, infarct-related artery, in-stent thrombosis, preprocedural TIMI flow 0, use of stent and DES, mechanical support, DAPT, multivessel disease, additional PCI, year of intervention, propensity score, and COVID positivity) (in-hospital death: OR [95% CI] = 0.75 [0.62–0.9], *p* = 0.003; 30-day death: HR [95% CI] = 0.74 [0.64–0.86], *p* < 0.0001).

## 4. Discussion

Our study is one of the largest reports on the association between smoking status and mortality in STEMI patients undergoing primary angioplasty, especially during the COVID pandemic. We found that active smokers had significantly better epicardial reperfusion and both in-hospital and 30-day survival compared with previous smokers and non-smokers. The association persisted even after correction for all baseline confounders, including the year of intervention, COVID positivity, and the propensity score.

Cigarette smoking is a well-known risk factor for coronary artery disease [5,36]. In fact, smokers were younger than non-smokers and were less likely to have additional established risk factors than non-smokers, suggesting the deleterious effect of smoking as a cause of myocardial infarction. However, despite the adjustment for all these confounders, smoking was still associated with a reduced mortality. While we cannot certainly exclude masked unmeasured confounders, it is possible that underlying biological differences in pathophysiology and response to the treatment in smokers versus non-smokers with STEMI could also have accounted, at least in part, for this paradoxical association.

Several studies conducted with thrombolysis have shown that smoking was associated with a lower mortality at both short- and long-term follow up [37,38,39]. Although, in most of these studies, the association between smoking and reduced mortality disappeared after correction for multiple confounders, some other studies observed a persistently lower mortality, even after adjustment. One of the explanations is the fact the smoking does not affect atherosclerotic plaque vulnerability, whereas it induces a hypercoagulation and prothrombotic state by endothelial dysfunction, increased platelet activation and aggregation, increased circulating levels of fibrinogen, and increased thrombin generation [40,41]. It has been shown, indeed, that components of cigarette smoke impair fibrin crosslinking [42]. Therefore, smoking may be predominantly thrombogenic and less likely atherogenic, making these patients more amenable for thrombolytic therapy and able to obtain more benefits from antiplatelet therapies. In fact, among patients reperfused with thrombolysis, smoking is associated with better epicardial [38] and myocardial reperfusion [43] compared with non-smokers.

Several studies have recently investigated the smoking paradox among patients undergoing mechanical reperfusion, with conflicting results. Redfors et al. [44] reported the prognostic impact of smoking among patients enrolled in CADILLAC trial. The authors observed a significantly lower mortality at 30-day and 1-year follow up in smokers compared with non-smokers. However, the difference disappeared after adjustment for all confounding factors. In another study by Steele et al. [45] including 3133 STEMI patients undergoing mechanical reperfusion, smoking was associated with a significantly increased mortality (hazard ratio 1.35 (95% CI 1.04–1.74)) compared with never smokers at 3 years after adjustment for differences in baseline variables. The risk for ex-smokers (hazard ratio 0.99 (0.76–1.28)) was similar to never smokers.

Opposite findings have been observed in the largest study so far conducted in primary PCI. Gupta et al., including more than 900.000 STEMI patients [46], found that smoking was associated with a significantly lower mortality, even after the adjustment for all baseline confounders. However, the mortality difference between smokers and non-smokers diminished substantially with increasing age and was no longer significant in nonagenarians with STEMI. These data suggest that the overall association of smoking with lower in-hospital mortality is driven mostly by younger age groups. However, given the lack of angiographic data, the authors were unable to account for the severity of CAD or to assess the procedural success of PCI. Another limitation is the very short-term follow up.

The present study is one of the largest on primary PCI, including more than 16,000 STEMI patients, and the first study conducted in the COVID era. In fact, the COVID-19 pandemic has been shown to increase mortality among STEMI patients by both direct and indirect effects, including a longer delay to presentation. We found that active smoking was associated with a significantly improved epicardial reperfusion (TIMI 3), especially as compared with non-smokers. Furthermore, active smokers had a significantly lower mortality as compared with non-smokers and previous smokers. Our results were confirmed in the sub analysis according to the year of intervention (COVID and pre-COVID era), age (older and younger than 75 years of age), and in patients with low or high propensity score values (below or upper the median value). The association between active smoking and angio-graphic and clinical outcome was confirmed after multivariate adjustment for all confounders, including the year of intervention and the propensity score.

A possible biological mechanism may explain that the paradoxically lower mortality among smokers treated by primary PCI is the potential different pathophysiology underlying the onset of infarction, mainly related among smokers to hypercoagulation and prothrombotic state by endothelial dysfunction, increased platelet activation and aggregation, increased circulating levels of fibrinogen, and increased thrombin generation [40,41], rather than atherosclerotic plaque vulnerability. This may also condition the response to several antithrombotic therapies.

In fact, a sub analysis of the HORIZONS-AMI trial also showed that, among STEMI patients undergoing pPCI for STEMI, bivalirudin monotherapy was associated with lower 30-day and 1-year mortality compared with unfractionated heparin plus glycoprotein IIb/IIIa inhibitors in smokers but not in non-smokers [47].

Furthermore, it has been suggested that smoking could affect responsiveness to antiplatelet therapies, mainly ADP antagonists. A sub analysis of the CLARITY-TIMI 28 trial, showed that clopidogrel reduced the rate of the primary end point (combined rates of occluded infarct-related artery or death and MI before angiography), especially among patients who smoked ≥10 cigarettes per day versus those who did not [48]. Similarly, in the CHARISMA trial clopidogrel reduced all-cause and cardiovascular mortality at 28 months in current smokers but not in non-smokers [19]. Similar impacts of smoking on the benefits from clopidogrel has been reported in other studies [18]. Pharmacokinetic and pharmacodynamics studies demonstrated that among patients treated with clopidogrel, smoking was associated with a greater inhibition of platelet aggregation, lower P2Y12 reaction units, and showed high platelet reactivity less often [20]. The induction of cytochrome P450 1A2 and 2B6 enzymes by cigarette smoking, both of which are involved in the hepatic biotransformation of clopidogrel to its active metabolite [49] has been identified as a possible explanation of the different response to clopidogrel between smokers versus non-smokers. Additional studies have been conducted on prasugrel, suggesting a similar effect of smoking status. A platelet-function sub study of the TRILOGY-ACS trial [21] showed that, among medically managed ACS patients randomly assigned to prasugrel or clopidogrel, smokers had lower P2Y12 reaction unit values at 6 months in both treatment groups compared with non-smokers. It has been demonstrated that nicotine is associated with higher P2Y12 receptor expression in human platelet lysates; this, therefore, could explain the observed effect of smoking on platelet inhibition [50]. These findings can contribute to understanding the observed significant reduction of ischemic outcomes with prasugrel versus clopidogrel among smokers [21].

These data on the differential clinical efficacy of antithrombotic therapies in smokers versus non-smokers could be a possible mechanistic explanation for the association of smoking with lower in-hospital mortality in patients undergoing pPCI for STEMI.

It must be emphasized that our findings and the overall concept of the smoking paradox should not erroneously interpreted as the beneficial effects from cigarette smoking. In fact, the lower prevalence of conventional risk factors among active smokers indirectly support the promotion of atherothrombosis, by active smoking, that led to STEMI at a younger age. The harmful effects of smoking have been largely proven, and these modest differences in short-term survival would likely be offset by the long-term mortality attributable to cigarette smoking. Therefore, all efforts should be carried out to strongly promote smoking cessation as a public health measure to reduce the burden of cardiovascular disease and its related mortality.

### Study Limitations

This study is limited by its retrospective design. It was conducted during a pandemic emergency, which was challenging and expected to encounter missing data. Cumulative smoking exposure in terms of number of pack years could not be quantified, and we were unable to study the association of the amount of smoking with outcomes. We were also unable to determine the time of smoking cessation for former smokers and neither did we assess infarct size. Moreover, even after statistical correction, the large differences in some strong prognostically relevant variables, particularly much younger age, do not allow for us to exclude that smoking is mostly a marker of STEMI at a younger age, where this risk factor is largely predominant as a direct prothrombotic cause of coronary occlusion. Therefore, the large differences in prognostically relevant baseline characteristics suggest prudence with regard to causal conclusions.

## 5. Conclusions

Our study showed that smoking is independently associated with improved epicardial re-canalization and lower in-hospital and 30-day mortality as compared with both previous smokers and non-smokers.

## Figures and Tables

**Figure 1 jcm-11-06722-f001:**
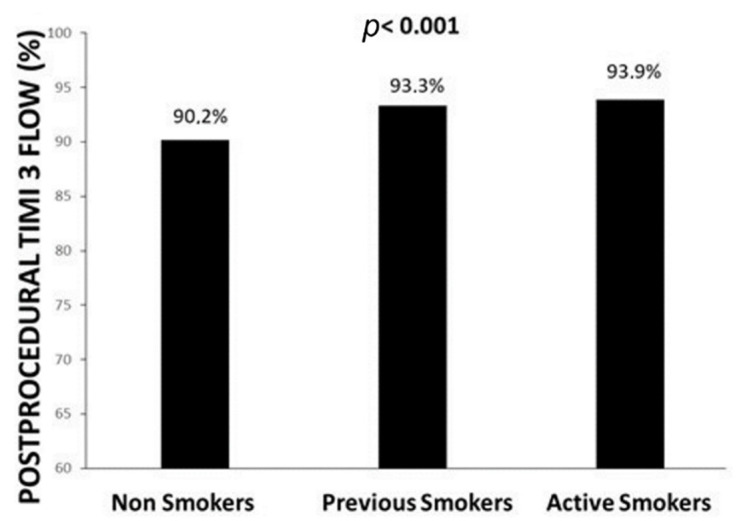
Bar graphs show the association between smoking status and postprocedural TIMI flow.

**Figure 2 jcm-11-06722-f002:**
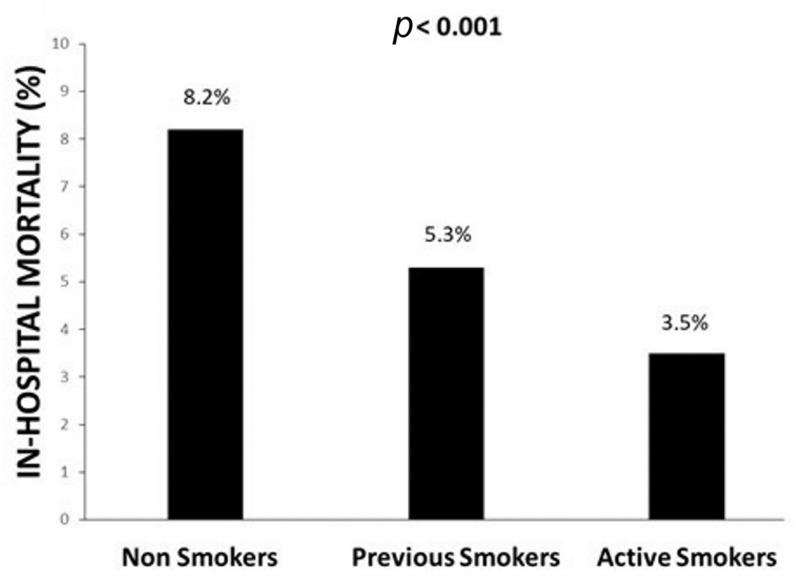
Bar graphs show the association between smoking status and in-hospital mortality.

**Figure 3 jcm-11-06722-f003:**
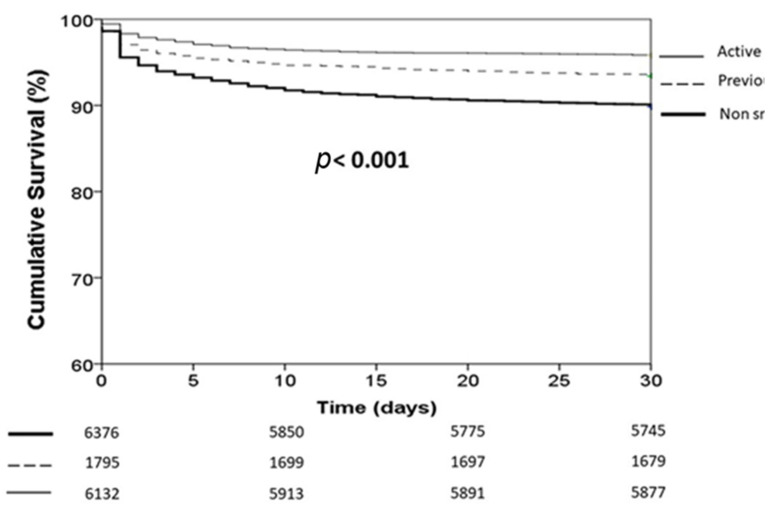
Kaplan–Meier survival curves of STEMI patients according to smoking status.

**Table 1 jcm-11-06722-t001:** Baseline demographic and clinical characteristics according to smoking status.

Variable	Active Smokers	Previous Smokers	Non-Smokers	*p* Value
	(*n* = 6819)	(*n* = 2099)	(*n* = 7165)	
Age (median, IQR)	58 (51–65)	67 (59–75)	67 (58–77)	<0.001
		(54–72)	(54–72)	
Age > 75 year—*n* (%)	410 (6.0)	533 (25.4)	2104 (29.4)	<0.001
Male gender—*n* (%)	5538 (81.2)	1773 (84.5)	4853 (67.7)	<0.001
Diabetes mellitus—*n* (%)	1324 (19.4)	502 (23.9)	1986 (27.7)	<0.001
Hypertension—*n* (%)	3252 (47.7)	1280 (61.0)	4281 (59.7)	<0.001
Hypercholesterolemia—*n* (%)	2579 (37.8)	1033 (49.2)	2741 (38.3)	<0.001
Family history of CAD—*n* (%)	1667 (24.4)	495 (23.6)	1136 (15.9)	< 0.001
Previous STEMI—*n* (%)	571 (8.4)	335 (16)	637 (8.9)	<0.001
Previous PCI—*n* (%)	722 (10.6)	432 (20.6)	839 (11.7)	<0.001
Previous CABG—*n* (%)	57 (0.8)	73 (3.5)	142 (2.0)	<0.001
Referral to primary PCI hospital				
Type				
Ambulance (from community)—*n* (%)	3328 (48.8)	1074 (51.2)	3336 (46.6)	<0.001
Direct access—*n* (%)	1820 (26.7)	512 (24.4)	2181 (30.4)	
Spoke—*n* (%)	1671 (24.5)	513 (24.4)	1648 (23.0)	
Time delays				
Ischemia time, median (25–75th)	190 (10–350)	208 (128–379)	221 (130–400)	<0.001
Total ischemia time				
<6 h—*n* (%)	5228 (76.7)	1550 (73.8)	5144 (71.8)	<0.001
6–12 h—*n* (%)	973 (14.3)	328((15.6)	1198 (16.7)	
12–24 h—*n* (%)	415 (6.1)	142 (6.8)	531 (7.4)	
>24 h—*n* (%)	2.3 (3.0)	79 (3.8)	292 (4.1)	
Total ischemia time > 12 h—*n* (%)	618 (9.1)	221 (10.5)	823 (11.5)	<0.001
Door-to-balloon time, median (25–75th)	40 (25–60)	38 (21–65)	40 (25–73)	<0.001
Door-to-balloon time				
<30 min—*n* (%)	2827 (41.5)	887 (42.3)	2719 (37.9)	<0.001
30–60 min—*n* (%)	2305 (33.8)	636 (30.3)	2318 (32.4)	
>60 min—*n* (%)	1687 (24.7)	576 (27.4)	2128 (29.7)	
Door-to-balloon time > 30 min (%)—*n* (%)	3992 (58.5)	1212 (57.7)	4446 (62.1)	<0.001
Clinical presentation				
Anterior STEMI—*n* (%)	2994 (43.9)	869 (41.4)	3583 (50.0)	<0.001
Out-of-hospital cardiac arrest—*n* (%)	374 (5.5)	91 (4.3)	491 (6.9)	<0.001
Cardiogenic shock—*n* (%)	408 (6.0)	157 (7.5)	603 (8.4)	<0.001
Rescue PCI for failed thrombolysis—*n* (%)	579 (8.5)	77 (3.7)	443 (6.2)	<0.001
In-hospital RASI therapy—*n* (%)	3864 (56.7)	1223 (58.3)	3810 (53.2)	<0.001
COVID positivity (%)	28 (0.4%)	16 (0.8%)	65 (0.9%)	<0.001

A Mann–Whitney test. CAD = coronary artery disease; STEMI = ST-segment elevation myocardial infarction; PCI = percutaneous coronary intervention; CABG 0 = coronary artery bypass graft.

**Table 2 jcm-11-06722-t002:** Angiographic and procedural characteristics.

	Active Smokers	Previous Smokers	Non-Smokers	*p* Value
	(*n* = 6819)	(*n* = 2099)	(*n* = 7165)	
Radial Access (%)	5289 (77.6)	1698 (80.9)	5281 (73.7)	<0.001
Culprit vessel				
Left main—*n* (%)	82 (1.2)	49 (2.3)	121 (1.7)	<0.001
Left anterior descending artery—*n* (%)	2976 (43.6)	852 (40.6)	3530 (49.3)	
Circumflex—*n* (%)	1064 (15.6)	362 (17.2)	924 (12.9)	
Right coronary artery—*n* (%)	2664 (39.1)	812 (38.7)	2525 (35.2)	
Anterolateral branch—*n* (%)	12 (0.2)	7 (0.3)	22 (0.3)	
SVG *n* (%)	20 (0.3)	17 (0.8)	42 (0.6)	
In-stent thrombosis—*n* (%)	238 (3.5)	127 (6.1)	267 (3.7)	<0.001
Multivessel disease—*n* (%)	3145 (46.1)	1062 (50.6)	3679 (51.3)	<0.001
Preprocedural TIMI 0 flow—*n* (%)	4632 (67.9)	1353 (64.5)	4746 (66.2)	0.007
Thrombectomy—*n* (%)	1112 (16.3)	342 (16.3)	1109 (15.5)	0.36
Stenting—*n* (%)	6391 (93.7)	1929 (91.9)	6447 (90)	<0.001
Drug-eluting stent—*n* (%)	6150 (90.2)	1858 (88.5)	6246 (87.2)	<0.001
Postprocedural TIMI 3 flow—*n* (%)	6402 (93.9)	1958 (93.3)	6461 (90.2)	<0.001
Gp IIb-IIIa inhibitors/cangrelor—*n* (%)	1404 (20.6)	441 (21.0)	1422 (19.8)	0.38
Bivalirudin—*n* (%)	18 (0.3)	9 (0.4)	25 (0.3)	0.44
Mechanical support—*n* (%)	164 (2.4.)	68 (3.2)	265 (3.7)	<0.001
Additional PCI				
During the index procedure—*n* (%)	627 (9.2)	260 (12.4)	689 (9.6)	0.001
Staged—*n* (%)	716 (10.5)	223 (10.6)	747 (10.4)	
DAPT therapy—*n* (%)	6769 (99.3)	2071 (98.7)	7065 (98.6)	<0.001

TIMI = thrombolysis in myocardial infarction; DAPT = dual antiplatelet therapy; RASI: renin-angiotensin system inhibitors.

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
