# Peer review of "Impact of Smoking Status on Mortality in STEMI Patients Undergoing Mechanical Reperfusion for STEMI: Insights from the ISACS–STEMI COVID-19 Registry"

_jcm, 2022, doi:10.3390/jcm11226722_

Round 1

Reviewer 1 Report

Dr. De Luca et al. reported the paradoxical lower in-hospital and 30-day mortality in active smokers compared to ex-smokers or non-smokers following STEMI, known as the smokers' paradox in the COVID era. The authors retrospectively investigated a large number of STEMI patients using a large-scale multicenter registry database and showed an interesting association between active smoking and lower mortality post-STEMI, which remained significant even after adjustment for multiple confounders; however, the study's retrospective nature, causal relationships are difficult to elucidate. Also, due to the lack of a clear mechanism underlying the observed association (all the discussion regarding the mechanism is based on speculation, but not on the provided data), the study should be considered hypothesis-generating.

Major comments

This reviewer could not see the supplemental tables and figures for review. 

Page 5 Line 209-212: Please provide the statistical method for propensity score technique more in detail. This reviewer could not find the statistical method in reference 22 that the author cited here. 

The authors adjusted for the below or above median propensity score for sensitivity analysis. However, this reviewer thinks that adjusting for whether the propensity score is higher or lower than the median may not be sufficient. Please consider propensity score matching analysis which can be extended to 3 groups comparison (Epidemiology. 2013 May;24(3):401-9. doi: 10.1097/EDE.0b013e318289dedf.). 

STEMI patients with active smoking were younger and less likely to have conventional cardiovascular risk factors compared to those with previous or non-smoking habits. This means that smoking habits can promote coronary atherosclerosis without these risk factors. More emphasis should be placed on the adverse effects of smoking in the discussion to avoid conveying the wrong messages.

Cardiogenic shock is less frequent in active smoking groups with less frequent use of mechanical support than in the other groups, which may indicate that the myocardial damage was less in the active smoking group. Can the authors provide the information regarding the infarct size (maximum CK increase, LVEF post-STEMI, etc.), which might have impacted the outcomes?

Minor comments

Page 4, Line 187: Please delete references (11-13) unrelated to the prior sentence.

Page 4, Line 198: Please delete the comma before the citation.

Page 4, Line 201-202: Please add a citation for the ARC definition.

Page 5, Line 215-218: The authors adjusted for the variables that differ among the three groups but not between the two groups.

Page 5, Line 237-238: How can the authors define the pre- and post-COVID era? Please describe it in the manuscript.

Page 11, Line 383: Please delete the repetitive word "data" here.

Author Response

Major comments

This reviewer could not see the supplemental tables and figures for review. 

REPLY 1) We are sorry for the inconvenient. Please find attached supplementary materials to the manuscript.

Page 5 Line 209-212: Please provide the statistical method for propensity score technique more in detail. This reviewer could not find the statistical method in reference 22 that the author cited here. 

REPLY 2) We thank the reviewer for the request. We now provide more details in the statistical section on the propensity score technique (page 6, lines 4-23).

The authors adjusted for the below or above median propensity score for sensitivity analysis. However, this reviewer thinks that adjusting for whether the propensity score is higher or lower than the median may not be sufficient. Please consider propensity score matching analysis which can be extended to 3 groups comparison (Epidemiology. 2013 May;24(3):401-9. doi: 10.1097/EDE.0b013e318289dedf.). 

REPLY 3) Being the aim of the current study to evaluate the prognostic impact of active smoking, we preferred to keep our propensity score analysis as it is, based on the differences between active smokers vs the rest of the population (page 6, lines 4-23).

STEMI patients with active smoking were younger and less likely to have conventional cardiovascular risk factors compared to those with previous or non-smoking habits. This means that smoking habits can promote coronary atherosclerosis without these risk factors. More emphasis should be placed on the adverse effects of smoking in the discussion to avoid conveying the wrong messages.

REPLY 4) We thank the reviewer for the comment. We now provide more emphasis on the adverse effects of smoking (page 14, lines 24-27).

Cardiogenic shock is less frequent in active smoking groups with less frequent use of mechanical support than in the other groups, which may indicate that the myocardial damage was less in the active smoking group. Can the authors provide the information regarding the infarct size (maximum CK increase, LVEF post-STEMI, etc.), which might have impacted the outcomes?

REPLY 5) We thank the reviewer for the suggestion. Unfortunately, data on infarct size were not collected and not available, as now stated in our limitations (page 14, line 38)

Minor comments

Page 4, Line 187: Please delete references (11-13) unrelated to the prior sentence.

REPLY 6) We have modified accordingly.

Page 4, Line 198: Please delete the comma before the citation.

REPLY 7) We have modified accordingly

Page 4, Line 201-202: Please add a citation for the ARC definition.

REPLY 8) We have added the citation on ARC definition (ref. 32)

Page 5, Line 215-218: The authors adjusted for the variables that differ among the three groups but not between the two groups.

REPLY 9) We thank the reviewer for the comment. We included in the adjustment all the variables significantly different between active smokers vs the rest of the population, that were the same variables significantly different among the three groups. This is clearly reported also on page 6, lines 28-31.

Page 5, Line 237-238: How can the authors define the pre- and post-COVID era? Please describe it in the manuscript.

REPLY 10)  We now provide more details on the definition of pre and post-COVID period.

Page 11, Line 383: Please delete the repetitive word "data" here.

REPLY 11) We have modified accordingly

Reviewer 2 Report

With great interest I have read the manuscript by Giuseppe De Luca et al regarding the "smoking paradox" in a cohort of STEMI patients undergoing PCI in the era of COVID-19 pandemic. 

Even though the strength of the study is limited by its retrospective design, one has to admit that the presentation is adequate and the number of participants seems large enough. After a quick search of the literature on the term "smoking paradox", I was not able to find studies with such a large number of patients. 

My main concern on the study is the big differences regarding the baseline characteristics, as one can see in Table 1. All the precipitating factors for the development of cardiovascular disease, such as diabetes, hypertension and family history of coronary artery disease seem to differ statistically among smoking and non-smoking groups, a fact that casts the results doubtful (the population is rather heterogenous). 

Author Response

My main concern on the study is the big differences regarding the baseline characteristics, as one can see in Table 1. All the precipitating factors for the development of cardiovascular disease, such as diabetes, hypertension and family history of coronary artery disease seem to differ statistically among smoking and non-smoking groups, a fact that casts the results doubtful (the population is rather heterogenous). 

REPLY 1) We agree with the reviewer’s comments. We have tried to overcome this limitation by the adjustment for all confounders, including the propensity score.

Round 2

Reviewer 1 Report

The authors properly responded to the previous comments from this reviewer. This reviewer has several minor comments below.

P value regarding the association between SARS-COV2 positivity and smoking status in elderly patients in the manuscript (Line 259) is different from the one in Figure 2S. Please check and correct the data.

Please put "=" between P and 0.018 in Figure 3S.

For Figures 3 and 7S-9S, please provide the number at risk in each figure.

In Line 281, HR for 30-day death may be a mistake for OR.

Author Response

We would like to thank the reviewer for these further comments and suggestions that have been helpful to improve and refine of our manuscript.

The authors properly responded to the previous comments from this reviewer. This reviewer has several minor comments below.

  • P value regarding the association between SARS-COV2 positivity and smoking status in elderly patients in the manuscript (Line 259) is different from the one in Figure 2S. Please check and correct the data.

REPLY 1. Thank you very much for your note. We have corrected it.

  • Please put "=" between P and 0.018 in Figure 3S.

REPLY 2. Thank you for your note. We have corrected it.

  • For Figures 3 and 7S-9S, please provide the number at risk in each figure.

REPLY 3. We have added the number at risk for Figure 3 and Figures 7S-9S

  • In Line 281, HR for 30-day death may be a mistake for OR.

REPLY 4. Hazard ratio is correct.

Reviewer 2 Report

The Authors took under consideration the Reviewer's concerns. As far as I can see, the propensity score seems to work well on the heterogeneity of the groups. I have no further comments to address.

Author Response

Thank you for the suggestions and comments